# Metacognitive Scales: Assessing Metacognitive Knowledge in Older Adults Using Everyday Life Scenarios

**DOI:** 10.3390/diagnostics12102410

**Published:** 2022-10-05

**Authors:** Grigoria Bampa, Despina Kouroglou, Panagiota Metallidou, Magdalini Tsolaki, Georgios Kougioumtzis, Georgia Papantoniou, Maria Sofologi, Despina Moraitou

**Affiliations:** 1Laboratory of Psychology, Section of Cognitive and Experimental Psychology, School of Psychology, Aristotle University of Thessaloniki, 54124 Thessaloniki, Greece; 2Laboratory of Neurodegenerative Diseases, Center of Interdisciplinary Research and Innovation (CIRIAUTH), Balkan Center, Buildings A & B, 57001 Thessaloniki, Greece; 3School of Medicine, Aristotle University of Thessaloniki, 54124 Thessaloniki, Greece; 4Greek Association of Alzheimer’s Disease and Related Disorders (GAADRD), 54643 Thessaloniki, Greece; 5Department of Turkish and Modern Asian Studies, National and Kapodistrian University of Athens, 15772 Athens, Greece; 6Laboratory of Psychology, Department of Early Childhood Education, School of Education, University of Ioannina, 45110 Ioannina, Greece; 7Institute of Humanities and Social Sciences, University Research Centre of Ioannina (URCI), 45110 Ioannina, Greece

**Keywords:** aging, attention, daily functioning, memory, metacognitive knowledge

## Abstract

The multidimensional effect of aging on cognition and its interference with daily functioning is well reported by many studies. Therefore, the ability to detect age-related cognitive changes is of great importance for older adults to help compensate for cognitive decline. For that, metacognition and its course of change across the lifespan of a person have attracted considerable scientific interest. The aim of the present study is to present three new self-report questionnaires, developed to measure older adults’ metacognitive knowledge for everyday memory (MKEM), metacognitive knowledge for everyday attention (MKEA), and metacognitive knowledge for everyday executive functions (MKEEFs). The questionnaires were tested for structural validity and reliability. A sample size of 171 community-dwelling adults of advancing age (97 females and 74 males) voluntarily participated in this study and their ages ranged from 50 to 82 years (mean = 59.32, SD = 7.39). Exploratory factor analysis using principal component analysis with varimax rotation was applied to examine structural validity. The results revealed a one-factor structure for the MKEM with high internal consistency (α = 0.88), a two-factor structure for the MKEA, that reflected “divided and shifted attention” (α = 0.74) and “concentration” (α = 0.75), and a two-factor structure for the MKEEFs that reflected “planning” (α = 0.70) and “inhibition” (α = 0.65). The variables created for each factor respectively showed significant positive correlations between each other.

## 1. Introduction

Metacognition describes the ability to monitor and regulate our cognitive processes. According to Flavell [1], metacognitive knowledge and metacognitive regulation comprise two of the core dimensions of metacognition. Specifically, metacognitive knowledge relates to general knowledge and beliefs a person has regarding how people learn and process information, but also to an individual’s knowledge and beliefs of one’s own cognitive skills. Metacognitive regulation refers to the ability of an individual to monitor their cognitive functions and to consciously reflect upon them when necessary [2]. As a person gets older, cognitive difficulties often appear and may interfere with daily functioning. Therefore, metacognitive skills are important for them to detect age-related cognitive changes. However, it is still an ongoing debate about which aspects of metacognition and to what extent are compromised in advanced age.

### 1.1. Metacognition and Aging

According to several studies, older adults tend to report overconfidence about their actual performance in cognitive tasks compared to younger adults [3,4,5,6]. Furthermore, in their study, Ross, Dodson, Edwards, Ackerman, and Ball [7] found that when older adults were asked to appraise their driving skills, the majority considered themselves as “good” or “excellent” drivers, despite the fact that during the last five years, it was more frequent for them to have been involved in an accident. These findings suggest that metacognitive beliefs and awareness of one’s own cognitive performance lack accuracy as a person ages. However, in their study, McGillivray and Castel [4] found that metacognitive predictions of older adults on a word-recall task became more accurate with task-related experience and feedback. Specifically, in their experiment, they requested older and younger adults to “bet” on the words they believed that they would later recall. Each word was paired with a number that indicated the word’s value. The goal was for the participants to collect as many points as possible. Their results showed that during the first trials, all participants overestimated their performance and the discrepancy between predictions and actual performance was greater for older adults than it was for younger adults. However, trial by trial, older adults not only became more accurate in their predictions, but they were as able as their younger opponents to strategically choose and learn the “high-valued” words to maximize their performance. The finding that older adults are able to incorporate feedback and task-related experience in order to minimize their initial overconfidence received further support from the results of Siegel and Castel [6]. However, their results showed that even though the discrepancy between older adults’ predictions and their actual performance decreased over trials, they remained overconfident in their overall predictions compared to younger adults.

In a recent chapter, McGillivray [8] provides a valuable overview of the available findings on metacognition in older adulthood. Aspects such as motivation, personal interest, and emotional valence seem to be important moderating factors for older adults’ cognitive efficiency. In particular, some studies have shown that when older adults are required to learn information about emotional valence, their memory performance was similar to that of their younger opponents [9]. Furthermore, regarding the effect of emotional content in information processing, it has been repeatedly reported that older adults tend to prioritize positive information and disregard negative information [8,10]. These results indicate that control processes in older adults present a selectivity effect in favor of positively valued stimuli. 

To sum up, research on metacognition and aging is still in progress. The available findings indicate that aging does not affect all dimensions of metacognition the same way [11]. Specifically, in older adults, while metacognitive control may remain relatively intact, metacognitive awareness and beliefs about one’s cognitive skills seem to be affected. Hence, if older adults have difficulties in accurately perceiving and monitoring their cognitive skills, it is reasonable to expect that their ability to compensate for age-related difficulties would be compromised [12]. To give a practical example, if one believes that one’s memory skills are relatively intact, one will have no reason to put in more effort when learning new information or to practice a memory strategy when trying to memorize a doctor’s appointment. Therefore, it is important to understand the ability of older adults to monitor their cognitive skills and to accurately detect any cognitive pitfalls. 

### 1.2. Age-Related Cognitive Changes

Abundant research findings have shown that various cognitive domains are vulnerable to age-related changes. Several terms and classification systems have been used to describe these aging effects. Discrimination between “crystallized” and “fluid” abilities has been widely used. Specifically, older adults show impairments in fluid abilities, which are related to efficiency and speed of information processing, problem solving and reasoning, while crystallized abilities, which are related to general knowledge, such as vocabulary and experience-based skills, remain intact or even improve as a person ages [13]. For the purpose of this study, the effects of cognitive aging are suggested to share a relationship with the following three major cognitive domains: memory, attention, and executive functions.

Memory loss is one of the primary complaints of older adults. However, not all aspects of memory decline as a person gets older. The types that are typically impaired are episodic memory (the ability to remember past events or experiences and their contextual details), prospective memory (memory for future intentions and tasks to be completed at a specific time and place), and working memory (the ability to online maintain and manipulate information for a short period of time) [14,15,16]. Similarly, aging does not equally affect all types of attention. Past studies have reported that older adults perform poorly in tasks that require divided or shifted attention [17,18], while their ability to maintain concentration for an extended period of time (sustained attention) does not show deterioration with aging [18]. However, although older adults efficiently maintain focus on stimuli of personal interest and value, they have difficulties suppressing distractions [19,20]. Deficits in executive functions (EFs) are also well reported, showing that all core dimensions of EFs; namely, inhibition (self-control), cognitive flexibility and set shifting (the ability of alternative thinking when obstacles appear), and planning (the ability to develop a roadmap that will enable the person to achieve a goal), do not remain intact in aging [17,21].

### 1.3. Existing Questionnaires on Metacognition and the Need for a New Tool

Existing questionnaires that are widely used for the assessment of metacognition in older adults, such as the Metamemory in Adulthood (MIA) [22], the Everyday Memory Questionnaire (EMQ) [23], or the Multifactorial Memory Questionnaire (MMQ) [24], focus solely on metamemory. The brief questionnaire on metacognition that was developed by Klusmann et al. [25] targets metacognitive beliefs about memory, as well as attention. However, the attention subscale of this questionnaire measures mainly sustained attention and disregards other aspects of attention, such as divided and shifted attention, which are typically affected by cognitive aging. In addition, the Cognitive Failures Questionnaire (CFQ) [26], even though it assesses subjective cognitive functioning in a variety of cognitive aspects (i.e., memory, orientation, control and inhibition, attention), it is deficit-oriented, and the answers are often given in a yes/no format. As a result, the retrieved information on how older adults perceive their cognitive functioning is still limited. 

Therefore, while the existing metacognitive questionnaires mainly target metamemory, the novelty of the presented questionnaires is that they assess older adults’ metacognitive knowledge in a variety of cognitively demanding situations, including attention and executive functions. 

### 1.4. Aim and Hypotheses of the Present Study

Following the overview of the available results regarding cognitive deficits in older adulthood and considering the limitations of the existing metacognitive tools, we developed three self-report questionnaires to measure metacognitive knowledge in adults of advanced age. Each questionnaire was developed to target a specific cognitive domain using a variety of everyday life scenarios (see Section 2.2 for a detailed description). The first questionnaire was developed to measure metacognitive knowledge for everyday memory (MKEM), the second was developed to measure metacognitive knowledge for everyday attention (MKEA), and the third was developed to measure metacognitive knowledge for everyday executive functions (MKEEFs). The purpose of this study was to investigate their psychometric properties in terms of factorial structure and reliability. It was expected that the MKEM consists of the following three theoretical factors with accepted internal reliability: (1) episodic memory, (2) prospective memory, and (3) working memory (Hypothesis 1). With regard to the MKEA, it was expected to reveal the following four theoretical factors with accepted internal reliability: (1) selective attention, (2) sustained attention, (3) divided attention, and (4) shifted attention (Hypothesis 2). With regard to the MKEEFs, it was expected to reveal the following three factors of accepted reliability, based on the respective theory: (1) inhibition, (2) planning, and (3) switching (Hypothesis 3). In addition, it was expected that since all three questionnaires were developed to measure metacognitive knowledge, their subscale factors would correlate positively with each other (Hypothesis 4). 

## 2. Materials and Methods

### 2.1. Participants and Procedure

To be eligible for this study, participants should be Greek native speakers, 50 years of age and above, and have a minimum education of 6 years. In addition, exclusion criteria for this study were as follows: medical history or current diagnosis of a neurological (i.e., dementia, diagnosed MCI, stroke, brain injury) and/or a psychiatric disorder (i.e., psychosis or clinical depression). A total of 171 participants (97 females and 74 males) were recruited through public workplaces and community centers. To calculate the necessary sample size for the presented study, “a rule of thumb” of ten participants per item was applied [27]. Since the MKEA had the most items (15 items), a sample size of 150 was estimated as the minimum number required. Hence, the recruited sample size was considered adequate. The age of participants ranged from 50 to 82 years (mean = 59.32, SD = 7.39) and their level of education varied from 6 to 29 years (mean = 13.15, SD = 4.86). 

The questionnaires were administered individually by the first and the second author (Grigoria Bampa and Despina Kouroglou) at a quiet place and properly equipped with a desk and a chair. The administration lasted approximately ten to fifteen minutes, depending on the participants’ pace. Participation in this study was voluntary and informed consent was obtained from all subjects involved in the study. 

### 2.2. Instruments

Metacognitive questionnaires 

Following the theoretical background in cognitive aging, three self-report questionnaires were developed to measure older adults’ metacognitive knowledge in a variety of everyday life situations. Specifically, the purpose of these scales was to assess how older adults perceive their cognitive skills in practical daily activities. As reported, existing metacognitive tools, although widely used and useful instruments, focus mainly on specific cognitive aspects (memory) and disregard others (dimensions of attention and executive functions); or there are deficits regarding providing limiting information about metacognitive knowledge in advanced age. Hence, the new questionnaires were developed to measure older adults’ metacognitive knowledge in terms of beliefs of efficacy. Furthermore, each questionnaire focuses on a specific cognitive domain and its main theoretical dimensions. 

For each scenario, participants were requested to estimate their degree of efficacy on a 4-point Likert scale, ranging from 1 = not at all to 4 = very well. Since the questionnaires refer to everyday life situations, efficacy in the described scenarios may vary under different circumstances and if that was the case, participants were instructed to consider their usual performance. Furthermore, if participants thought that a described scenario does not precisely correspond to a situation from their daily life, they were instructed to imagine themselves in the presented situation and try to estimate their most likely degree of efficacy.

#### 2.2.1. Metacognitive Knowledge for Everyday Memory (MKEM)

The MKEM questionnaire consists of twelve items developed to measure metacognitive knowledge related to episodic memory (3 items; example item: “Imagine that someone (i.e., a friend, son/daughter, etc.) asks you how you had spent the day before. How well do you manage to remember what you had done the previous day?”), prospective memory (3 items; example item: “Imagine that you are watching TV. A member of your family asked you to call him/ her when the news starts. How well do you manage to remember to call him/ her when the news starts?”) and working memory (3 items; example item: “Imagine that you ask for an address. You are looking for a pen to write it down. How well do you manage to remember the address until you find the pen?”).

#### 2.2.2. Metacognitive Knowledge for Everyday Attention (MKEA)

The MKEA is a 15-item self-report questionnaire developed to measure metacognitive knowledge concerning the following four aspects of attention: selective attention (3 items; example item: “How well do you manage to concentrate on a task you need to complete without getting distracted by a problem you are dealing with (work or personal life related)?”), sustained attention (4 items; example item: “Imagine that you are reading a large text (at least 2 pages long). How well do you manage to remain concentrated until you finish it?”), divided attention (4 items; example item: How well do you manage to have a conversation while you are cooking or driving?”), and shifted attention (4 items; example item: “Imagine that you are having a conversation, suddenly another person enters the room and asks you something irrelevant, for example, the time. How well do you manage to answer the question and then continue the conversation you had from the point you left it?”). 

#### 2.2.3. Metacognitive Knowledge for Everyday Executive Functions (MKEEFs)

The MKEEFs questionnaire consists of twelve items developed to measure metacognitive knowledge concerning the following three aspects of executive functions: inhibition (3 items; example item: “Imagine that you are at a dinner gathering with family and/or friends. How well do you manage to stop yourself from drinking or eating something that your doctor has forbidden you?”), planning (3 items; example item: “Imagine that you want to attend a theater performance or to go on a trip with the bus. How well do you manage to call and book a ticket for the time and date that works better for you?”) and switching (3 items; example item: “Imagine that you are on the phone and suddenly someone is knocking at your door. How well do you manage to turn off the phone and open the door?”).

### 2.3. Statistical Analysis

The statistical analysis was conducted using SPSS Version 27.0. Exploratory factor analysis (EFA) was applied to test structural validity using principal component analysis (PCA) for component extraction. For all three scales, the varimax rotation method was applied; a minimum eigenvalue of 1 was set as the extraction criterion, and the criterion for factor loading was set at ≥0.40. To determine the number of the extracted factors for each scale, the following criteria were considered: the eigenvalues that estimated the percentage of explained variance to be >1, the scree plots, the Kaiser–Meyer–Olkin (KMO) that estimates the sample’s adequacy (values above 0.5 are considered adequate), and Bartlett’s test of sphericity. The rotated component matrix was used to explore factor loadings on the factors. New variables were created by adding the scores of the items’ loading for each factor of every questionnaire. The scale’s reliability was assessed by means of internal consistency using Cronbach’s alpha coefficient. Convergent validity using Pearson’s correlations was tested to assess the relationships between the newly developed variables. 

## 3. Results

### 3.1. Exploratory Factor Analysis of the MKEM Scale

Data from 167 participants (four participants were excluded due to missing data) were analyzed to explore the structure of the MKEM questionnaire. The overall KMO was 0.90, indicating that the recruited sample size was adequate for factor analysis. Bartlett’s test of sphericity was significant with χ^2^(66) = 688.844, *p* < 0.001. Factor analysis revealed a one-factor solution, which accounted for 42.95% of the total variance (eigenvalue = 5.15). The unidimensional structure was further supported by the scree plot, which clearly indicated one factor. Each one of the 12 items presented moderate loadings (0.5–0.7) on the factor. Regarding the scale’s reliability, Cronbach’s alpha coefficient for all 12 items was 0.88, indicating almost excellent internal consistency. Table 1 displays the results of the MKEM questionnaire.

### 3.2. Exploratory Factor Analysis of the MKEA Scale

Data from 165 participants (six participants were excluded due to missing data) were analyzed to explore the structure of the MKEA questionnaire. The overall KMO was 0.87 and Bartlett’s test of sphericity was significant with χ^2^(105) = 754.178, *p* < 0.001. The initial factor analysis revealed a four-factor solution, which accounted for 60.05% of the total variance. The scree plot displayed a flattening curve when more than two factors were used. In addition, a close examination of the factor loadings revealed that three items (4, 10, 15) had moderate loadings (<0.6) on two factors. 

In the next step, a three-factor solution was tested. The second factor analysis accounted for 53.17% of the total variance. The overall KMO was 0.87. Bartlett’s test of sphericity was significant with χ^2^(105) = 754.178, *p* < 0.001. The scree plot displayed a flattening curve when more than two factors were used. Again, an examination of factor loadings revealed that three items (4, 6, 15) had moderate loadings (<0.6) on two factors. Therefore, the three items were removed, and the same analysis was re-ran. This time, the three-factor solution accounted for 54.24% of the total variance. The overall KMO was 0.82 and Bartlett’s test of sphericity was significant with χ^2^(66) = 481.272, *p* < 0.001. Again, the scree plot presented a flattening curve when more than two factors were used. All items had moderate to strong loadings (0.4–0.9) and no item had loadings on more than one factor. Based on this model, the first factor (eigenvalue = 3.99) was loaded by five items, the second (eigenvalue = 1.45) by five items, and the third (eigenvalue = 1.07) by two items. 

Since the third factor was loaded only by two items and the scree plots from the previous steps clearly indicated two factors, the analysis was re-ran for a two-factor solution for the twelve items. The two factors explained 45.34% of the total variance. The overall KMO was 0.82 and Bartlett’s test of sphericity was significant with χ^2^(66) = 481.272, *p* < 0.001. The scree plot presented the same image as in the previous step. An examination of factor loadings revealed that each one of the twelve remaining items had moderate to strong loading (0.5–0.8) on one of the two factors. Specifically, the first factor (eigenvalue = 3.99) was loaded by seven items, and the second (eigenvalue = 1.45) by five items. 

After close examination of all the analysis steps that were performed, a two-factor solution for the twelve remaining items was considered the best-fitting structure for the MKEA questionnaire. Cronbach’s alpha coefficient for the seven items of the first factor, labeled as “divided and shifted attention” was 0.75, and for the five items of the second, labeled as “concentration”, it was 0.74, indicating adequate internal consistency for each factor. The results for the MKEA questionnaire are displayed in Table 2.

### 3.3. Exploratory Factor Analysis of the MKEEFs Scale

Data from 169 participants (two participants were excluded due to missing values) were analyzed to explore the structure of the MKEEFs. From the initial analysis, a three-factor solution emerged, which accounted for 53.42% of the total variance. The overall KMO was 0.82. Bartlett’s test of sphericity was significant with χ^2^(66) = 451.494, *p* < 0.001. However, the scree plot presented a flattening curve when more than two factors were used. In addition, an examination of factor loadings revealed that one item had loadings on two factors. Thus, this item was removed, and the analysis was re-ran.

A two-factor solution emerged after item removal, and it accounted for 45.45% of the total variance. The overall KMO was 0.80. Bartlett’s test of sphericity was significant with χ^2^(55) = 384.862, *p* < 0.001. Again, factor loadings revealed that one item had loadings (<0.5) on two factors. This item was removed, and the analysis was re-ran.

The next factor analysis revealed a two-factor solution, which accounted for 46.42% of the total variance. The overall KMO was 0.79 and Bartlett’s test of sphericity was significant with χ^2^(45) = 313.525, *p* < 0.001. Each of the ten remaining items presented moderate to strong loadings (0.4–0.8) on one of the two factors and the two factors were loaded by five items each. This was considered the best-fitting structural model for the MKEEF questionnaires. Cronbach’s alpha coefficient for the five items of the first factor, labeled “planning”, was 0.70, and for the five items of the second factor, labeled as “inhibition”, it was 0.65, indicating an adequate and a relatively low internal consistency for each factor, respectively. The results for the MKEEF questionnaire are displayed in Table 3.

### 3.4. Correlations among Variables Representing the Factor Subscales 

The three questionnaires were tested for convergent validity. A new variable was created for each factor of each questionnaire by adding the scores of the items that were loaded on each factor. The results showed significant positive correlations between the new variables, confirming that all three questionnaires are related to the theoretical construct they measure. However, Pearson correlations between MKEA: divided and shifted attention and the two factors of the MKEEFs scale were very high (see Table 4), indicating that they may describe the same theoretical construct. Therefore, we decided to run a new EFA, including all the items from the MKEEF scale and the items from MKEA: divided and shifted attention, to test whether new, more integrated factors would emerge. However, the results were similar to the aforementioned ones; three factors were revealed, including one for divided and shifted attention and two for planning and inhibition, respectively. Only three items of the MKEΕFs were found to load on the attention factor. Moreover, the reliability of the inhibition factor remained the same (low). Thus, we decided to accept the initial factorial models. Table 4 displays the correlations.

## 4. Discussion

Cognitive aging affects various cognitive domains, and it may interfere with daily functioning. Are older people aware of these changes? Are they able to detect these changes in their daily life? What do they believe about their cognitive skills in daily activities? To elucidate these questions, we developed three self-report questionnaires using a variety of scenarios from older adults’ everyday life. The questionnaires were administered to a Greek population of community-dwelling adults of advanced age, and they were tested for their psychometric properties regarding the scale’s validity (factorial and convergent) and reliability (internal consistency). 

The MKEM questionnaire was created to assess metacognitive knowledge regarding different aspects of memory, such as working memory, episodic memory, and prospective memory; therefore, three theoretical factors were expected to be identified. Based on the results, the MKEM questionnaire showed good psychometric properties. However, the hypothesis (H1) regarding its factorial structure was not confirmed. The results showed that participants did not discriminate between these types, but rather perceived memory as a one-dimension function, that is, everyday memory. This finding indicates that older adults have poor metacognitive knowledge that pertains to different aspects of memory, and thus poor insight into how memory works. Furthermore, if older adults are not able to discriminate between different aspects of memory, and thus between situations with different cognitive demands (it is one thing to retain a short amount of information for a few seconds, i.e., working memory, but another thing to remember next week’s doctor’s appointment, i.e., prospective memory), it is of no surprise that they do not effectively use encoding strategies [8,28,29,30]. Similarly, studies that have tested the effects of metacognitive trainings in the elderly have shown that they were able to use more efficiently mnemonic strategies and self-regulatory processes when they obtained a better understanding of how memory operates [29,31]. Therefore, it would be interesting for future research to examine if older adults can discriminate between different dimensions of memory after participating in metacognitive training. 

The results also showed good psychometric properties for the MKEA. According to the second hypothesis, a four-factor solution was expected for the MKEA scale. The results indicated that participants perceived attention as a two-dimensional function, contrary to our initial hypothesis (H2) of a four-factorial structure (selective, sustained, divided, and shifted attention). The items that identified the first factor reflect situations in which divided or shifted attention is required, indicating that participants perceived these aspects of attention as one dimensional; thus, it was labeled as “divided and shifted attention”. The second factor was perceived as the ability to focus on a specific task and ignore distractions; thus, it was labeled as “concentration”. According to these results, it seems that older adults discriminate attention derived by external factors, such as distractions or the number of stimuli they need to pay attention to, rather than by inner regulatory processes, such as maintaining, shifting, or dividing attentional resources. If this is the case, a major question arises regarding their ability to allocate and control attentional resources in response to task demands. However, since research about metacognition in advanced age has exclusively favored metamemory, it is difficult to make specific hypotheses regarding meta-attention in older adults. Certainly, future studies need to also enlighten this pathway and metacognitive training for older adults should also incorporate meta-attention in their context. 

According to the third hypothesis (H3), a structure of three theoretical factors was expected for the MKEEF scale (switching, inhibition, and planning). This hypothesis was not confirmed and a two-factor solution was revealed as the most fitting for this scale. Switching between different tasks or mindsets was not identified as a third distinct factor, but as part of the “planning” factor. A possible explanation is that the related items were formulated in a very similar way (for example, item 2 “Imagine that you are on the phone and suddenly someone is knocking at your door. How well do you manage to turn off the phone and open the door?” was created to describe switching processes and item 4 “Imagine that you want to attend a theater performance or to go on a trip with the bus. How well do you manage to call and book a ticket for the time and date that works better for you?” was created to describe the planning process). The items that identified the first factor describe situations where a person needs to organize and execute a series of tasks; thus, it was labeled as “planning”. The items that identified the second factor describe situations where a person is required to regulate his/her behavior and not act impulsively; thus, it was labeled as “inhibition”. The psychometric properties for the MKEEF scale were adequate to satisfactory.

The fourth hypothesis (H4) of the study was confirmed, and all three scales are related in terms of the specific constructs they measure, all of which constitute metacognitive knowledge related to everyday life tasks. The strong relationships between attention and executive functions have received considerable scientific interest and it is an ongoing debate whether attention is a core component of executive functions or a distinct and fundamental cognitive function. Several studies lean towards the first hypothesis. Specifically, it has been proposed that attention serves as a top-down attentional control mechanism [32], playing a fundamental role in goal-directed behavior and multitasking [33,34]. Based on this approach, attention that is referred to in the literature as attention control or executive attention [34,35] is described as the ability to manage and regulate attentional processes towards the relevant stimuli or task sets, while suppressing distractions and temptations until the goal/ task is finished. Therefore, the strong correlations that were found between the divided and shifted attention and the planning and inhibition factors of MKEEFs dovetail with the attentional control model, suggesting that attention is a component of an executive system, but still a distinct dimension.

### Limitations

There are several limitations that must be reported. Further examination of the psychometric properties of the newly developed questionnaires is required. More specifically, the present study did not test for convergent validity between the MKEM, MKEA, and MKEEFs and other scales that measure metacognition, such as the EMQ and MFQ. Furthermore, divergent validity also needs to be tested to establish that MKEM, MKEA, and MKEEFs are different from questionnaires that measure similar, but distinct, traits. Another limitation concerns the sample of this study. The sample of the current study was comprised of community-dwelling adults of advanced age who did not undergo any extended neuropsychological assessment. Hence, it is possible that some of them may be undiagnosed MCI cases. Since the relationship between MCI and metacognition is still under investigation, it is not easy to define how or whether this affected the study’s results. Moreover, a more representative sample of older adults is needed. Finally, regarding the specific content of the questionnaires, two items may be considered outdated (item 1 from MKEM scale: “Imagine that you want to call someone. You read the phone number to memorize it. How well do you manage to remember the phone number without looking at it again?”. Nowadays, phone numbers are usually saved as phone contacts, so there is no need to memorize and dial a phone number) or confusing (item 3 from MKEEFs scale: “Imagine that you are on the phone and suddenly someone is knocking at your door. How well do you manage to turn off the phone and open the door?”. Several participants mentioned that they can do both together and they do not understand why the described situation should cause any trouble). These items should be replaced.

## 5. Conclusions

Despite the limitations that certainly need to be addressed by future research, the present study also has strengths, supporting its contribution to the scientific and clinical community. The results revealed good psychometric properties for all three metacognitive scales; thus, the presented questionnaires could enrich the array of scientific tools for assessing metacognitive knowledge in adults of advanced age. While the existing metacognitive questionnaires mainly target metamemory, the presented questionnaires assess older adults’ metacognitive knowledge in a variety of cognitively demanding situations, including attention and executive functions. Regarding the theoretical structure of the questionnaires, the initial hypotheses were not confirmed. However, this should be of no surprise for several reasons. First, it is very challenging to represent a single cognitive aspect using practical daily paradigms because performance in real-life activities is usually the product of the collaboration between multiple cognitive processes. Furthermore, it is possible that participants do not obtain metacognitive knowledge to differentiate between specific dimensions of cognitive functions. Future research should further investigate this hypothesis and explore whether aging affects the perception of different cognitive dimensions. Another reason that certain dimensions were not identified may be due to the formulation of some items. Regarding their clinical relevance, they can be used by clinicians to gain valuable information regarding patients’ beliefs of efficacy in a variety of daily situations. Furthermore, comparing their answers to their actual cognitive performance, as measured by neuropsychological tests, or to information received from caregivers can provide estimations about patients’ awareness of their deficits.

## Figures and Tables

**Table 1 diagnostics-12-02410-t001:** Exploratory factor analysis for the MKEM questionnaire.

Items	Everyday Memory
Imagine that you want to call someone. You read the phone number to memorize it. How well do you manage to remember the phone number without looking at it again?	0.62
Imagine that someone (i.e., a friend, son/daughter, etc.) asks you what you did yesterday. How well do you manage to remember what you did the previous day?	0.68
Imagine that you are watching TV and in fifteen minutes, you need to remember to turn off the boiler. In order to do so, you set a reminder, for instance your alarm. How well do you manage to remember what it is you need to do when you hear the alarm?	0.54
Imagine that you are on your way, and you ask a passer-by for directions in order to find the address you are looking for. How well do you manage to remember the directions you received until you find your destination?	0.60
Imagine that are calling to arrange a doctor’s appointment for the next week. How well do you manage to remember the appointment (by memory)?	0.74
Imagine that you want to tell a story that you read earlier in a book or in a newspaper. How well do you manage to remember details of that story, such as names, place and time?	0.63
Imagine that at the end of the week, you need to pay a bill. How well do you manage to remember it (by memory)?	0.64
Imagine that you ask for an address. You are looking for a pen to write it down. How well do you manage to remember the address until you find the pen?	0.71
Imagine that you are asking a member of your family how he/she is going to spend the day. How well do you manage to remember what he/ she told you without asking again?	0.67
Imagine that you are watching TV. A member of your family has asked you to call him/ her when the news start. How well do you manage to remember to call him/ her when the news starts?	0.67
Imagine that you are at the supermarket’s counter check and the cashier tells you the amount you need to pay. In the meantime, you are looking for your wallet. How well do you manage to remember the amount you need to pay until you open your wallet?	0.61
How well do you manage to remember details (i.e., names, place, time) from a conversation you had earlier?	0.73
**Eigenvalue**	**5.15**
**% of variance**	**42.95**
**Reliability**	**0.88**

**Table 2 diagnostics-12-02410-t002:** Exploratory factor analysis for the MKEA questionnaire.

Items	Divided and Shifted Attention	Concentration
How well do you manage to follow a conversation in which more than two people participate?	0.71	
Imagine that you arrive at the train or bus station much earlier before departure time. As a result, you are walking around the station to spend your spare time. How well do you manage to not lose your route announcement?	0.69	
Imagine that you are at the bank, and you are waiting for your number to appear on the announcement table. How well do you manage to stay focused so that you do not lose your turn when your number appears?	0.69	
How well do you manage to have a conversation while you are cooking or driving?	0.59	
Imagine that you are having a conversation and suddenly, another person enters the room and asks you something irrelevant, for example the time. How well do you manage to answer the question and then continue the conversation you had from the point you left it?	0.55	
Imagine that you are waiting for the streetlights to turn green while at the same time you are talking on your phone. How well do you manage to initiate crossing the street when the lights turn green?	0.53	
Imagine that you are making your groceries list and suddenly, the phone rings. You stop to answer the phone. After turning off the phone, how well do you manage to continue your list from the point you left it?	0.51	
How well do you manage to concentrate on a task you need to complete without getting distracted by a problem you are dealing with (work or personal life related)?		0.76
How well do you manage to concentrate on reading a text (i.e., book, magazine, newspaper) when there is noise (i.e., noise from the street or kids playing)?		0.70
Imagine that you are reading a large text (at least 2 pages long). How well do you manage to remain concentrated until the end?		0.68
How well do you manage to watch a TV program and talk with someone at the same time?		0.65
How well do you manage to read something while listening to music?		0.61
**Eigenvalue**	**3.99**	**1.45**
**% of variance**	**33.26%**	**12.08%**
**Reliability**	**0.74**	**0.75**

**Table 3 diagnostics-12-02410-t003:** Exploratory factor analysis for the MKEEM questionnaire.

Items	Planning	Inhibition
Imagine that you have planned to go on a walk with a friend, but it starts raining. How well do you manage to think of an alternative plan considering the weather (i.e., sit in a cafeteria)?	0.80	
Imagine that you are on the phone and suddenly, someone is knocking at your door. How well do you manage to turn off the phone and open the door?	0.75	
Imagine that you are having a conversation with someone and suddenly, the phone rings. How well do you manage to pause the conversation and answer the phone?	0.66	
Imagine that you want to attend a theater performance or to go on a trip on a bus. How well do you manage to call and book a ticket for the time and date that works better for you?	0.65	
Imagine that you have planned with friends or family to go out for dinner. You enter a restaurant but there are no free tables. How well do you manage to think of an alternative place?	0.41	
Imagine that it is the beginning of the month, and you just received your salary/pension. How well do you manage to handle the money to cover the month’s expenses?		0.75
Imagine that you are in a room where you need to keep quiet (i.e., at a doctor’s office or at a library). How well do you manage to not talk loudly?		0.60
Imagine that you are at a dinner gathering with family and/or friends. How well do you manage to stop yourself from drinking or eating something that your doctor has forbidden you?		0.63
Imagine that you are at the bank, and someone steals your turn. How well do you manage to not react in a rude way?		0.61
Imagine that you decide to go on a trip this weekend. How well do you manage to pack your suitcase with the things you are going to need?		0.44
**Eigenvalue**	**3.50**	**1.50**
**% of variance**	**31.81%**	**13.64%**
**Reliability**	**0.70**	**0.65**

**Table 4 diagnostics-12-02410-t004:** Correlations among the retained components of the MKEM, MKEA, and MKEEFs.

	MKEM: Everyday Memory	MKEA: Divided and Shifted Attention	MKEA: Concentration	MKEEFs: Planning
MKEA: divided and shifted attention	0.65 **			
MKEA: concentration	0.55 **	0.48 **		
MKEEFs: planning	0.63 **	0.81 **	0.74 **	
MKEEFs: inhibition	0.64 **	0.81 **	0.63 **	0.59 **

** Correlation is significant at the 0.01 level (2-tailed).

## Data Availability

The data of this study are available at DOI:10.17632/xzwv9wzgz2.1.

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
