# Peer review of "Metacognitive Scales: Assessing Metacognitive Knowledge in Older Adults Using Everyday Life Scenarios"

_diagnostics, 2022, doi:10.3390/diagnostics12102410_

Round 1

Reviewer 1 Report

This is a very good piece.

Author Response

We would like to express our appreciation for taking the time and effort necessary to review the manuscript. Your encouraging comment is greatly appreciated.

Kind regards, 

Grigoria Bampa

Reviewer 2 Report

The article is scientifically very well structured

A fundamental analysis and theme 

We consider that we have nothing to add

Author Response

We would like to express our appreciation for taking the time and effort necessary to review the manuscript. Your encouraging comments are greatly appreciated. Also, the manuscript will be carefully revised by a professional to improve English language, as you suggested.

Kind regards, 

Grigoria Bampa

Reviewer 3 Report

§  What were the inclusion and exclusion criteria?

§  Age range was 50-82 years. Isn’t this wide gap? Moreover, cognitive dysfunction generally starts after 60-65 years.

§  Novelty of the study must be clearly mentioned at the end of the Introduction section.

§  Some old references like 1st, 2nd etc. can be updated with recent ones.

Author Response

 Dear Reviewer,

I would like to express our appreciation for your comments and suggestions. They helped us to improve our manuscript. I hope that I have replied sufficiently to all your suggestions.

Point 1: What were the inclusion and exclusion criteria?

Response 1: I reformed the section Participants so that the inclusion and exclusion criteria are presented more clearly (lines 159-163).

“To be eligible for this study, participants should be Greek native speakers, 50 years of age and above, and have a minimum education of six years. In addition, exclusion criteria for this study were: medical history or current diagnosis of a neurological (i.e., dementia, diagnosed MCI, stroke, brain injury) and/or a psychiatric disorder (i.e., psychosis or clinical depression).”

Point 2:  Age range was 50-82 years. Isn’t this wide gap? Moreover, cognitive dysfunction generally starts after 60-65 years.

Response 2: It is indeed a wide age range. Cognitive decline mostly starts after 60 years, however, it is well reported that it can also begin in midlife. This is also what we have experienced after years of collaboration with “Alzheimer Hellas” conducting research and clinical work. Therefore, we used a wide age range because we wanted to include them all.  

Point 3: Novelty of the study must be clearly mentioned at the end of the Introduction section.

Response 3: I described the novelty of the current study at the end of the Introduction section as you suggested (lines 133-136).

“Therefore, while the existing metacognitive questionnaires mainly target metamemory, the novelty of the presented questionnaires is that they assess older adults’ metacognitive knowledge in a variety of cognitively demanding situations including attention and executive functions.”

Point 4: Some old references like 1st, 2nd etc. can be updated with recent ones.

Response 4: Thank you for your suggestion. Unfortunately, I cannot change these old references. These are the researchers who developed the related theories.

Kind regards,

Grigoria Bampa

Reviewer 4 Report

In the manuscript “Metacognitive scales: Assessing metacognitive knowledge in older adults using everyday life scenarios”, the authors developed  three self-report questionnaires to investigate older adults’ metacognitive knowledge for everyday memory (MKEM), metacognitive knowledge for everyday attention (MKEA), metacognitive knowledge for everyday executive functions (MKEEFs) respectively,  and found that a one-factor structure for the MKEM,  a two-factor structure for the MKEA (“Divided & Shifted Attention” and  “Concentration”), and a two-factor structure for the MKEEFs (“Planning” and “Inhibition”). The questionnaires developed in this study are a great contribution to clinical research and diagnosis. The world’s population of the older adult (aged 60 years and over) will grow year by year, and will be double (2.1 billion) by 2050. Such studies and the results are very helpful in maintaining the health of the cognition in older adults and in the prevention, diagnosis, and treatment of the diseases related.

One minor comment:

If we analyze the data based on age groups, say 50yr-59yr, 60yr-69yr, 70yr up or by gender, would  the results remain the same?

Author Response

We would like to express our appreciation for taking the time and effort necessary to review the manuscript. Your encouraging comments are greatly appreciated. Your suggestion to analyze the data by age group or gender is very interesting for future research. Unfortunately, the current sample size is not sufficient to conduct this kind of statistical analysis. Furthermore, the manuscript is under revision for English language by a professional to improve grammar and readability. 

Kind regards, 

Grigoria Bampa

Round 2

Reviewer 3 Report

Authors have responded to the comments.